# Improving LLM Video Understanding with 16 Frames Per Second

Yixuan Li [* 1]   Changli Tang [* 1]   Jimin Zhuang [1]   Yudong Yang [1]   Guangzhi Sun [2]   Wei Li [2]   Zejun Ma [2]
Chao Zhang [1]

## Abstract

Human vision is dynamic and continuous. However, in video understanding with multimodal large language models (LLMs), existing methods primarily rely on static features extracted from images sampled at a fixed low frame rate of frame-per-second (FPS) $\leqslant 2$, leading to critical visual information loss. In this paper, we introduce F-16, the first multimodal LLM designed for high-frame-rate video understanding. By increasing the frame rate to 16 FPS and compressing visual tokens within each 1-second clip, F-16 efficiently captures dynamic visual features while preserving key semantic information. Experimental results demonstrate that higher frame rates considerably enhance video understanding across multiple benchmarks, providing a new approach to improving video LLMs beyond scaling model size or training data. F-16 achieves state-of-the-art performance among 7-billion-parameter video LLMs on both general and fine-grained video understanding benchmarks, such as Video-MME and TemporalBench. Furthermore, F-16 excels in complex spatiotemporal tasks, including high-speed sports analysis (*e.g.*, basketball, football, gymnastics, and diving), outperforming SOTA proprietary visual models like GPT-4o and Gemini-1.5-pro. Additionally, we introduce a novel decoding method for F-16 that enables highly efficient low-frame-rate inference without requiring model retraining. We will release the source code, model checkpoints, and data at https://github.com/bytedance/F-16.

## 1. Introduction

Large language models (LLMs) have demonstrated exceptional performance across many natural language processing tasks, with some even nearing human-level performance (OpenAI et al., 2024; Dubey et al., 2024; Touvron et al., 2023; Du et al., 2022; Bai et al., 2023). The impressive ability of LLMs to understand, generate, and reason with text has sparked significant interest among researchers, drawing attention from both academia and industry to expand their capabilities to multimodal understanding.

In video understanding, videos contain both slowly changing elements, such as backgrounds and scenes, and rapidly changing, fleeting details, such as body movements and micro-expressions. Given the high frame count in videos, processing every frame is computationally expensive. As a result, existing video LLMs (Li et al., 2024; Zhang et al., 2024b; Wang et al., 2024; Lin et al., 2024b; Cheng et al., 2024; Tang et al., 2024; Yao et al., 2024) primarily focus on slowly changing elements, typically employing low-frame-rate sampling or selecting a fixed number of frames, effectively treating every 1-second video clip as a few images. However, this approach has a critical limitation: while low-frame-rate sampling like 1 frame-per-second (FPS) may be sufficient for capturing a video's overall theme and context, it fails to preserve rapidly changing visual cues, leading to information loss. This significantly hinders the model's ability to comprehend dynamic scenes, thereby limiting its overall video understanding capabilities.

To address these limitations, this paper introduces F-16, a video LLM designed for more human-like video perception by processing coherent video frames at a higher frame rate of 16 FPS. A key challenge in high-frame-rate video understanding is the increased visual token sequence length and redundant information across consecutive frames. To mitigate this, we propose a visual-text aligner that not only transforms visual features into text-like tokens suitable for the LLM backbone but also compresses redundant intra-frame and inter-frame information within each 1-second video clip. To leverage the strengths of existing high-performance image encoders, we adopt a 3-layer multi-layer perception (MLP) as the aligner, extending a pre-trained image LLM to process the dynamic features in 16 FPS videos while

---

[*]Equal contribution  [1]Tsinghua University [2]ByteDance. Correspondence to: Chao Zhang <cz277@tsinghua.edu.cn>.

*Proceedings of the $42^{nd}$ International Conference on Machine Learning*, Vancouver, Canada. PMLR 267, 2025. Copyright 2025 by the author(s).

preserving rich semantic features from each static frame. Experimental results demonstrate that F-16 achieves state-of-the-art (SOTA) performance among models of similar sizes and remains competitive with much larger models on general video question-answering (QA) benchmarks, including Video-MME (Fu et al., 2024), MLVU (Zhou et al., 2024), LongVideoBench (Wu et al., 2024), and NeXT-QA (Xiao et al., 2021), as well as fine-grained video understanding benchmarks like TemporalBench (Cai et al., 2024) and MotionBench (Hong et al., 2025). The superior performance of F-16 highlights the advantage of reducing visual information loss through 16 FPS sampling. Additionally, after fine-tuning on high-speed sports tasks such as basketball, football, gymnastics, and diving, F-16 significantly outperforms SOTA proprietary models like GPT-4o and Gemini-1.5-Pro. Finally, we introduce a training-free variable-frame-rate decoding method, allowing F-16 to adapt to lower-frame-rate scenarios without increasing test-time computational costs.

Our main contributions are summarized as follows:

- **The first high-frame-rate video LLM**: We developed F-16, the first video LLM capable of perceiving videos at 16 FPS. F-16 supports videos up to 110 seconds, processing 1,760 visual frames per video, and achieves SOTA performance on multiple video understanding benchmarks among 7-billion (B)-parameter video LLMs.

- **High-frame-rate sports video benchmarking**: To evaluate video LLM comprehension in high-frame-rate scenarios, we curated a dataset of sports videos that require fine-grained temporal understanding. F-16 significantly outperforms GPT-4o and Gemini-1.5-Pro on these tasks.

- **Efficient variable-frame-rate decoding**: We propose a variable-frame-rate decoding method that enables F-16, trained at 16 FPS, to be seamlessly applied to scenarios suitable for low-frame-rate videos, reducing computational cost without hurting the performance.

## 2. Related Work

### 2.1. From Image LLM to Video LLM

Most image LLMs adopt the approach of connecting an image encoder to an LLM via modality adapters, achieving remarkable success. LLaVA (Liu et al., 2024b;a) applies instruction tuning (Wei et al., 2022), enabling zero-shot image understanding. BLIP-2 (Li et al., 2023) integrates a frozen image encoder with an LLM using Q-Former to bridge the modalities. InternVL (Chen et al., 2023) further enhances accuracy by scaling up the visual encoder for more precise image representations. InternVL 2.5 (Chen et al., 2024) delves into model scaling and achieves better test-time performance.

Regarding video understanding, recent studies typically sample at a low frame rate or select a fixed number of frames, treating them as separate images with orders. In many approaches, each frame is first encoded using a pre-trained image encoder, then mapped to the text space via a modality aligner, and finally fed into an LLM backbone for response generation. Video-LLaVA (Lin et al., 2024a) uniformly samples 8 frames from a video, processes each frame independently through an image encoder, and generates video tokens that are then passed to the LLM. LLaVA-OneVision (Li et al., 2024) is designed to handle single images, multiple images, and videos. For videos, it samples frames at 1 FPS and reduces the number of video tokens using bilinear interpolation. Building on a similar architecture, Zhang et al. develops a strong video-understanding LLM with synthetic data, but still limits sampling to 1 FPS. Qwen2-VL (Wang et al., 2024) increases the frame rate to 2 FPS and employs a rotary position embedding to enhance temporal modelling. Meanwhile, VideoLLaMA 2 (Cheng et al., 2024) and video-SALMONN 2 (Tang et al., 2024) incorporate full audio information to support video understanding. However, VideoLLaMA 2 only processes 16 frames from each video, while video-SALMONN 2 uses no more than 30 frames.

Low frame rate sampling can result in critical visual information loss, particularly in videos with rapidly changing scenes, intricate details, or fast motion. Additionally, if keyframes are missed, yet the model is trained on labels that rely on keyframe information, it may struggle to align its predictions with the expected content, potentially leading to hallucinations and degraded performance. To address these challenges, Koala (Tan et al., 2024), Frame Voyager (Yu et al., 2024b), and KeyVideoLLM (Liang et al., 2024) employ intelligent sampling strategies to extract key information more effectively. Meanwhile, MA-LMM (He et al., 2024) and VideoStreaming (Qian et al., 2024) explore denser video sampling to enhance model performance.

### 2.2. Temporal Input Compression in Video LLM

More frames must be sampled to enable high-frame-rate video understanding, making temporal information compression between adjacent frames essential for efficient computation. A common approach aggregates image features extracted from each frame to compress redundant information effectively. RLT (Rafailov et al., 2024) reduces the total number of video tokens by dropping redundant patches, identified through frame differentials. Video-LaVIT (Jin et al., 2024) integrates keyframes and motion vectors as input and trains a motion encoder to enhance motion understanding. Espresso (Yu et al., 2024a) employs specialized spatial and temporal Q-poolers and compressors to eliminate redundant information. NVILA (Liu et al., 2024c) follows a "scale-then-compress" paradigm, first scaling up both frame resolution and frame count, then applying spatial linear compression and temporal averaging to reduce the

total number of visual tokens. Notably, these approaches mostly focus on processing long videos efficiently rather than processing videos with high frame rates.

# 3. Methods

## 3.1. Model Architecture

The overall architecture of F-16 is illustrated in Fig. 1. F-16 takes a video and a text prompt as the inputs and generates textual responses accordingly.

For the input video, frames are first sampled at a high frame rate, indicating the sampled frames are adjacent to each other and they may be visually very similar. Denote $\mathbf{F}_i$ as the $i$ th frame sampled from the video and $n$ as the total number of sampled frames. Each frame is encoded by the pre-trained image encoder $\mathrm{Enc}(\cdot)$ as

$$\mathbf{Z}_i = \mathrm{Enc}(\mathbf{F}_i), \ \ 0 \le i < n. \tag{1}$$

Here, $\mathbf{Z}_i \in \mathcal{R}^{p \times d}$ is the visual features of $\mathbf{F}_i$ output by the image encoder, where $p$ is the number of patches and $d$ is the output dimension of the image encoder.

Given the high similarity between adjacent frames, we aim to compress image features within a local time window in the visual feature space, preserving overall visual semantics while effectively capturing the dynamic features introduced by temporal changes across frames. Previous studies (Liu et al., 2024b;c) suggest that linear transformations outperform nonlinear ones in mapping image features to the LLM input space, as they better retain the semantic information extracted by the image encoder. Building on these insights, we design our high-frame-rate aligner using a 3-layer MLP, which consists of two linear layers with a GELU (Hendrycks & Gimpel, 2016) activation function in between.

Let $w$ be the number of frames in a processing window, and let $\mathbf{Z}_{jw}, \mathbf{Z}_{jw+1}, ..., \mathbf{Z}_{(j+1)w-1}$ represent the visual features of these $w$ frames in the $j$ th window, and $\mathbf{P}(\cdot)$ and $\mathbf{Q}(\cdot)$ as the first and second linear layers of the high-frame-rate aligner, respectively. To construct the input to the aligner, the visual features of all frames within the processing window are first concatenated along the feature dimension as:

$$\mathbf{Z}_j^{\mathrm{cat}} = \mathrm{Concat}(\mathbf{Z}_{jw}, \mathbf{Z}_{jw+1}, \ldots, \mathbf{Z}_{(j+1)w-1}), \tag{2}$$

where $\mathbf{Z}_j^{\mathrm{cat}} \in \mathcal{R}^{p \times wd}$ is the combined visual features of the $j$ th window.

Next, the high-frame-rate aligner maps $\mathbf{Z}_j^{\mathrm{cat}}$ to the input dimension of the LLM, denoted as $h$. Specifically, the first linear layer $\mathbf{P}(\cdot)$ maps the features from the dimension of $wd$ to $wh$, while the second linear layer $\mathbf{Q}(\cdot)$ maps from $wh$ to $h$. Therefore, the output vectors $\tilde{\mathbf{H}}_j \in \mathcal{R}^{p \times h}$ of the high-frame-rate aligner for the $j$ th window can be obtained

using Eqns. (3) and (4) as:

$$\tilde{\mathbf{H}}_j = \mathbf{Q}(\mathrm{GELU}(\mathbf{P}(\mathbf{Z}_j^{\mathrm{cat}}))). \tag{3}$$

To further compress the visual tokens, a spatial $2 \times 2$ max pooling function $\mathrm{Max2DPool}(\cdot)$ is employed after the aligner, as shown in Eqn. (4). Specifically, $\tilde{\mathbf{H}}_j$ is first resized to $\mathcal{R}^{\sqrt{p} \times \sqrt{p} \times h}$, then $\mathrm{Max2DPool}(\cdot)$ is applied on the first and second dimension, which results in about 4 times reduction of the total number of visual tokens. Note that

$$\mathbf{H}_j = \mathrm{Max2DPool}(\tilde{\mathbf{H}}_j), \tag{4}$$

where $\mathbf{H}_j \in \mathcal{R}^{\lfloor \frac{\sqrt{p}}{2} \rfloor^2 \times h}$ are the visual tokens of the $j$ th processing window and are fed to the backbone LLM.

Finally, visual tokens of all non-overlapped processing windows form the final visual token sequence $\mathbf{H}$ of the whole video. The backbone LLM is required to generate a textual response $\hat{\mathbf{Y}}$ given the user's text instructions $\mathbf{I}$ and the visual token sequence $\mathbf{H}$ by

$$\hat{\mathbf{Y}} = \arg\max_{\mathbf{Y}} P(\mathbf{Y}|\mathbf{I}, \mathbf{H}). \tag{5}$$

## 3.2. Building F-16 by Extending Image LLM

Video LLMs are typically built on well-trained image LLMs to leverage their pre-trained visual perception abilities. However, a key challenge arises as shown in Fig. 1: the dimensions of the high-frame-rate aligner's modules $\mathbf{P}(\cdot)$ and $\mathbf{Q}(\cdot)$ do not match those of the single-image aligner in the image LLM when extending an image LLM to video LLM. This mismatch prevents direct parameter initialization using the pre-trained image LLM, as illustrated in Fig. 2. To address this, we adopt the block matrix decomposition approach, breaking down F-16's high-dimensional modality aligner into smaller sub-matrices that can be initialized using parameters from the image LLM, ensuring a smoother transfer of pre-trained knowledge.

Specifically, F-16 extends a pre-trained image LLM with LLaVA-OneVision structure (Li et al., 2024), which has a single-frame aligner with two linear layers as shown in Fig. 2(a). The aligner takes single-image features as input and outputs the corresponding visual tokens. Denote its first and second linear layers as $\mathbf{A}(\cdot)$ and $\mathbf{B}(\cdot)$, and their weight matrices and bias vectors as $\mathbf{W_A} \in \mathcal{R}^{d \times h}$, $\mathbf{W_B} \in \mathcal{R}^{h \times h}$, $\mathbf{b_A} \in \mathcal{R}^h$, and $\mathbf{b_B} \in \mathcal{R}^h$, respectively. Similarly, denote the weight matrices and bias vectors of the two linear layers $\mathbf{P}(\cdot)$ and $\mathbf{Q}(\cdot)$ of the high-frame-rate aligner in F-16 as $\mathbf{W_P} \in \mathcal{R}^{wd \times wh}$, $\mathbf{W_Q} \in \mathcal{R}^{wh \times h}$, $\mathbf{b_P} \in \mathcal{R}^{wh}$, and $\mathbf{b_Q} \in$

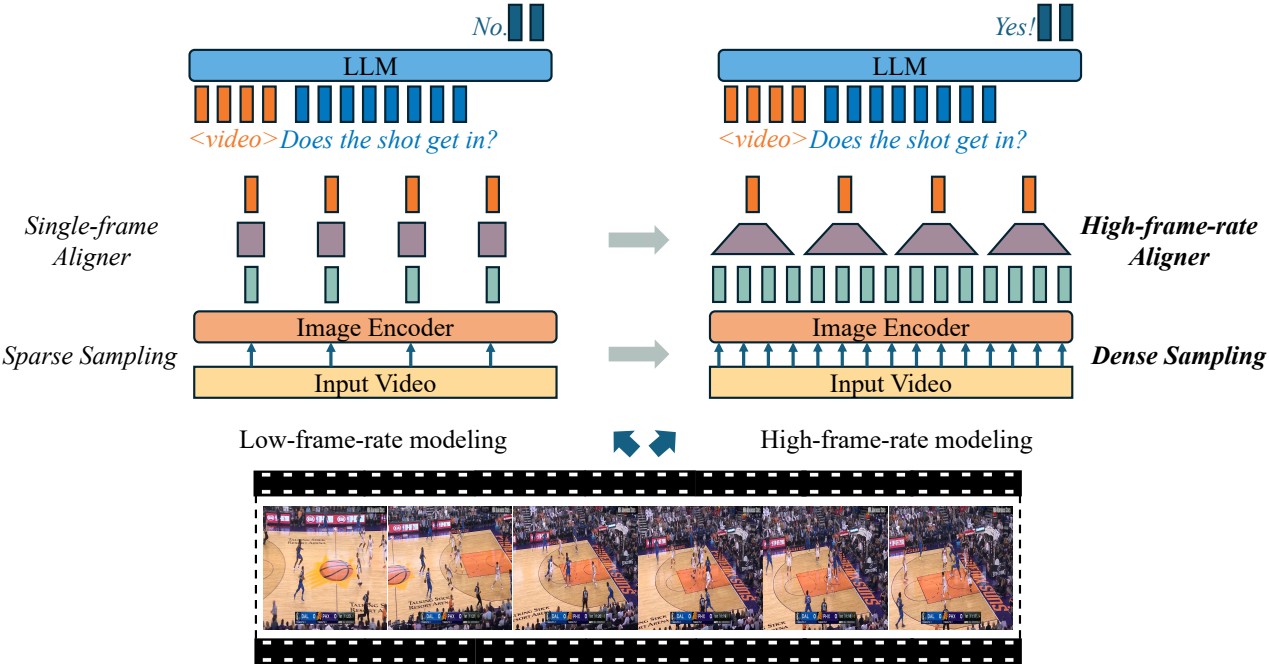

*Figure 1.* The overall architecture of F-16. Compared to the classical structure of low-frame-rate video LLMs, F-16 samples more frames with a higher frame rate of 16 FPS and uses a high-frame-rate aligner, which not only preserves the overall visual semantics but also extracts the dynamic features from the changes across frames, without introducing more visual tokens.

$\mathcal{R}^h$. F-16's parameters can be initialized using

$$\mathbf{W_P} = \begin{pmatrix} \mathbf{W_A} & \mathbf{O} & \cdots & \mathbf{O} \\ \mathbf{O} & \mathbf{W_A} & \cdots & \mathbf{O} \\ \vdots & \vdots & \ddots & \vdots \\ \mathbf{O} & \mathbf{O} & \cdots & \mathbf{W_A} \end{pmatrix}, \tag{6}$$

$$\mathbf{b_P} = \begin{pmatrix} \mathbf{b_A} \\ \mathbf{b_A} \\ \vdots \\ \mathbf{b_A} \end{pmatrix}, \mathbf{W_Q} = \frac{1}{w} \begin{pmatrix} \mathbf{W_B} \\ \mathbf{W_B} \\ \vdots \\ \mathbf{W_B} \end{pmatrix}, \mathbf{b_Q} = \mathbf{b_B}, \tag{7}$$

where $\mathbf{O}$ is a zero matrix. This initialization leads to the fact that F-16's aligner's output is the average of the representations of each frame in its initial state, since Eqn. (3) can be expanded using Eqns. (8)-(9) as:

$$\mathbf{P}(\mathbf{Z}_j^{\text{cat}}) = \begin{pmatrix} \mathbf{W_A} & \mathbf{O} & \vdots \\ \mathbf{O} & \ddots & \mathbf{O} \\ \cdots & \mathbf{O} & \mathbf{W_A} \end{pmatrix} \begin{pmatrix} \mathbf{Z}_{jw} \\ \mathbf{Z}_{jw+1} \\ \vdots \\ \mathbf{Z}_{(j+1)w-1} \end{pmatrix} + \begin{pmatrix} \mathbf{b_A} \\ \mathbf{b_A} \\ \vdots \\ \mathbf{b_A} \end{pmatrix}$$

$$= \begin{pmatrix} \mathbf{A}(\mathbf{Z}_{jw}) \\ \mathbf{A}(\mathbf{Z}_{jw+1}) \\ \vdots \\ \mathbf{A}(\mathbf{Z}_{(j+1)w+1}) \end{pmatrix}, \tag{8}$$

$$\tilde{\mathbf{H}}_j = \mathbf{Q}(\text{GELU}(\mathbf{P}(\mathbf{Z}_j^{\text{cat}})))$$

$$= \frac{1}{w} \begin{pmatrix} \mathbf{W_B} \\ \mathbf{W_B} \\ \vdots \\ \mathbf{W_B} \end{pmatrix}^{\text{T}} \begin{pmatrix} \text{GELU}(\mathbf{A}(\mathbf{Z}_{jw})) \\ \text{GELU}(\mathbf{A}(\mathbf{Z}_{jw+1})) \\ \vdots \\ \text{GELU}(\mathbf{A}(\mathbf{Z}_{(j+1)w+1})) \end{pmatrix} + \mathbf{b_B}$$

$$= \frac{1}{w} \sum_{k=0}^{w-1} \mathbf{B}(\text{GELU}(\mathbf{A}(\mathbf{Z}_{jw+k}))). \tag{9}$$

To improve training stability, in our implementation, we set the off-diagonal elements in the $\mathbf{W_P}$ to random noise initialized with Kaiming Uniform (He et al., 2015), rather than all zeros.

### 3.3. Variable-Frame-Rate Decoding

Encoding each frame with an image encoder significantly increases the computational load, making inference slower compared to low-frame-rate models. This becomes a major bottleneck for real-time processing. To address this, we introduce a variable-frame-rate decoding method, allowing F-16 to perform low-frame-rate inference, thereby reducing computational costs while maintaining strong performance.

At test-time, assume that we want to reduce the frame rate by a factor of $k$, compared to the frame rate used for training. The processing window can be correspondingly reduced to $w/k$. To meet the input dimension of the high-frame-rate

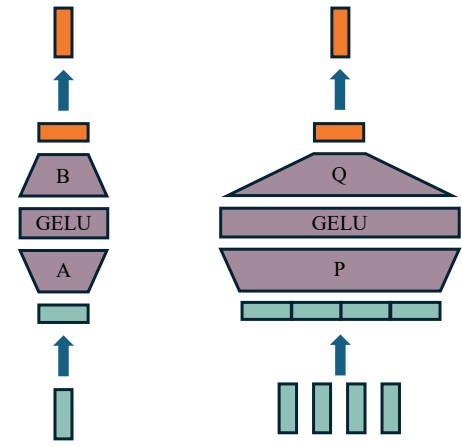

(a) Single-frame Aligner    (b) High-frame-rate Aligner

*Figure 2.* The comparison between (a) single-frame aligner and (b) high-frame-rate aligner. With more input frames, the high-frame-rate aligner concatenates the frames and uses linear layers $\mathbf{P}(\cdot)$ and $\mathbf{Q}(\cdot)$ that have larger input dimensions to compress and encode the frames to visual tokens.

aligner, the features of each frame need to be repeated $k$ times. Under this condition, in the $j$ th processing window, Eqn. (2) can be expressed as follows:

$$\mathbf{Z}'_{jw/k} = \text{Concat}(\underbrace{\mathbf{Z}_{jw/k}, \mathbf{Z}_{jw/k}, \ldots, \mathbf{Z}_{jw/k}}_{k \text{ times}}),$$

$$\mathbf{Z}^{\text{cat}}_j = \text{Concat}(\mathbf{Z}'_{jw/k}, \mathbf{Z}'_{jw/k+1}, ..., \mathbf{Z}'_{(j+1)w/k-1}).$$

Therefore, $\mathbf{Z}^{\text{cat}}_j \in \mathcal{R}^{p \times wd}$, which still meets the input dimension requirements of the high-frame-rate aligner.

## 4. Experimental Setup

### 4.1. Model Specifications

F-16 utilizes the LLaVA-OV model of LLaVA-OneVision (Li et al., 2024) with 7B parameters as based image LLM, which uses SigLIP (Zhai et al., 2023) as the visual encoder and Qwen2-7B (Yang et al., 2024) as the backbone LLM. Besides, F-16 samples frames from the video at 16 FPS as input, allowing for a maximum of $16 \times 110 = 1760$ frames. For videos longer than 110 seconds, F-16 uniformly samples 1760 frames from the video as input. The width $w$ of the processing window is set equal to the FPS as $w = 16$.

### 4.2. Data Specifications

The training data of general videos are the same as LLaVA-Video (Zhang et al., 2024b), including LLaVA-Video-178K (Zhang et al., 2024b), LLaVA-Hound (Zhang et al., 2024a), NExT-QA (Xiao et al., 2021), ActivityNet-QA (Yu et al., 2019) and PerceptionTest (Patraucean et al., 2024).

Besides generic video understanding, we also fine-tune the model on high-speed sports videos. Videos for gymnastics, diving, basketball, and football are collected for further tuning, where FineGym (Shao et al., 2020), Diving48 (Li et al., 2018), SoccerNet (Giancola et al., 2018), and NBA video clips are used respectively. To demonstrate the importance of high-frame-rate modeling in sports videos, the visualization for a video randomly sampled from Diving48 is shown in Appendix D.

Regarding the FineGym (Shao et al., 2020) data for gymnastics understanding, we sample 90% clips as the training set while the remaining 10% as the test set and ensure that the duration of videos in the training and test sets is balanced. Video captioning and open-ended QA of gymnastics are trained, and open-ended QA is evaluated. Accuracy is the used evaluation metric.

Regarding the Diving48 (Li et al., 2018) data for diving understanding, we use its official data split. In addition, the diving actions are split into four phases including takeoff, somersault, twist, and flight, following the original paper. Formatted captions describing all the phases are used to train models. The mean accuracy of predicting actions of the four phases is used as the evaluation metric.

The "Ball Action" subset of SoccerNet (Giancola et al., 2018) is used for football understanding. Counting the times of ball passes in the video is our focus task, since it is essential to use high frame-rate information. We segment the videos into 10-second clips and count the number of passes in each segment to provide training labels. Accuracy serves as the evaluation metric.

Regarding the NBA data, we collected NBA matches from 2024/11/13 to 2024/11/25, with a total number of 276 matches. We focus on the task of whether the ball goes into the net, which requires high-frame-rate information. We evaluated the performance using the test set of NSVA (Wu et al., 2022) using F1 score. We manually annotated video captions of 10,000 NBA video clips. These video captions focus on details of players' movements on the field, the movement of the ball, and other aspects of movement, providing the model with high-frame-rate data annotations. Examples of these sports data can be found in Appendix A.

### 4.3. Training Specifications

For general video training, the high-frame-rate aligner and the LLM are updated, while the image encoder stays frozen. F-16 is trained for 1 epoch on the training data using 128 H100 GPUs, with a learning rate set to $2 \times 10^{-5}$.

For further tuning the model on high-speed sports data, LoRA (Hu et al., 2022) is adapted to the LLM and serves as the only trainable module in this stage. The rank and the scaling factor of LoRA are set to 128 and 2.0, respectively. We fine-tune F-16 using 64 H100 GPUs for 5 epochs, with

*Table 1.* Comparison of video QA results of different video LLMs. The FPS or maximum number of frames sampled by the model is listed in the table. We evaluate the models on Video-MME, VideoVista (VST), TemporalBench (TPB), MotionBench (MB), NeXT-QA (NQA), MLVU, and LongVideoBench (LVB) for a comprehensive performance comparison. %Accuracy is the evaluation metric, except for TPB, which uses %Multiple Binary Accuracy (Cai et al., 2024) for evaluation. F-16 achieves SOTA on Video-MME, NQA, TPB, and MB among all 7B models and offers competitive results compared to SOTA proprietary models like GPT-4o and Gemini-1.5-Pro.

| Model | FPS / #Frame | Video-MME | | | | NQA ↑ | TPB ↑ | MB ↑ | VST ↑ | MLVU ↑ | LVB ↑ |
|---|---|---|---|---|---|---|---|---|---|---|---|
| | | Avg.↑ | S↑ | M↑ | L↑ | Test | Short | Val. | Avg. | m-Avg. | Val. |
| GPT-4o | 1FPS / 384 | 71.9 | 80.0 | 70.3 | 65.3 | - | 38.0 | 33.0 | 78.3 | 54.5 | 66.7 |
| Gemini-1.5-Pro | 1FPS | 75.0 | 81.7 | 74.3 | 67.4 | - | 26.6 | 51.0 | - | - | 64.0 |
| Qwen2-VL-7B | 2FPS / 768 | 63.3 | - | - | - | - | 24.7 | 52.0 | **75.6** | - | 55.6 |
| VideoLLaMA2-7B | 16 | 47.9 | - | - | - | - | - | - | 60.5 | 48.4 | - |
| VideoChat2-HD-7B | 16 | 45.3 | - | - | - | 79.5 | - | - | - | 37.4 | 39.3 |
| LLaVA-OV-7B | 32 | 58.2 | - | - | - | 79.4 | 21.2 | - | 73.0 | 64.7 | 56.3 |
| MiniCPM-V2.6-8B | 64 | 60.9 | 71.3 | 59.4 | 51.8 | - | 21.4 | 52.0 | - | - | 54.9 |
| LLaVA-Video-7B | 64 | 63.3 | - | - | - | 83.2 | 22.9 | - | - | **70.8** | **58.2** |
| NVILA-7B | 1024 | 64.2 | 75.7 | 62.2 | **54.8** | 82.2 | - | - | - | 70.1 | 57.7 |
| F-16-7B (ours) | 16FPS / 1760 | **65.0** | **78.9** | **63.2** | 52.8 | **84.1** | **37.2** | **54.5** | 74.4 | 70.3 | 57.6 |

a learning rate set to $2 \times 10^{-5}$.

## 5. Experimental Results

### 5.1. General Video Understanding

Results of the general video understanding are presented in Table 1. F-16 achieves the SOTA results among the 7B models in several video QA benchmarks, including not only the general video QA benchmarks Video-MME and NeXT-QA, but also the fine-grained video understanding benchmarks TemporalBench and MotionBench.

On the Video-MME Short, Medium, and NeXT-QA datasets—each designed for short video understanding—our model surpasses the previous 7B SOTA model by 3.2%, 1.0%, and 0.9% in accuracy, highlighting its strong performance on short videos. For benchmarks evaluating long video understanding, such as Video-MME Long, LongVideoBench, and MLVU, the challenge is greater due to sparser frame sampling, causing frames within the processing window to exhibit more significant variations. This increases the difficulty for the modality aligner to effectively encode temporal changes within the limited token representation. As a result, F-16 experiences a slight performance drop compared to LLaVA-Video-7B (Zhang et al., 2024b), which is trained on the same video dataset.

Results on TemporalBench and MotionBench which focus on motion and temporal understanding, show a significant positive effect of high-frame-rate modeling. F-16 gains an improvement of 13.5% and 2.5% respectively on the two benchmarks compared with the existing 7B SOTA models and also achieves competitive performance of SOTA

commercial models like Gemini-1.5-Pro and GPT-4o.

### 5.2. High-Speed Sports Video Understanding

In this section, we explore the high-speed sports video understanding of our models. We use the sports data of gymnastics, diving, basketball, and football to fine-tune our general video understanding models trained using FPS = 1 and FPS = 16. GPT-4o and Gemini-1.5-Pro are also evaluated by uniformly sampled 120 frames as its input, to provide an upper bound on the existing Video LLMs on these tasks. Note that both models cannot recognize gymnastics and diving actions, as these two scenarios require much expertise. The detailed results are shown in Table 2.

*Table 2.* Results of high-speed sports understanding. F-16 with the high-frame-rate aligner outperforms the low-frame-rate aligner version in all sports tasks. We also evaluate GPT-4o and Gemini-1.5-Pro on the NBA and SoccerNet QAs, which do not require the knowledge from the in-domain training set to answer the questions.

| Model | Gym | Diving | NBA | Soccer |
|---|---|---|---|---|
| | %Acc.↑ | %Acc.↑ | %F1↑ | %Acc.↑ |
| GPT-4o | - | - | 76.7 | 36.8 |
| Gemini-1.5-Pro | - | - | 80.6 | 43.1 |
| F-16 - FPS = 1 | 48.5 | 76.0 | 87.1 | 55.4 |
| F-16 - FPS = 16 | **64.1** | **86.5** | **92.9** | **57.7** |

The results demonstrate that our high-frame-rate model outperforms both the low-frame-rate model and commercial models across all four sports tasks, highlighting the importance of temporal perception. Among these tasks, high-frame-rate modelling shows the most significant advantage in gymnastics and diving, improving accuracy by over 15%

and 10%, respectively. This is likely because accurately distinguishing movements requires capturing the full motion sequence, rather than relying on sparse frames. For example, in gymnastics, determining whether a movement is clockwise or counterclockwise can be difficult with only sparsely sampled frames. Similarly, in diving, assessing the degree of rotation or twist is challenging without a continuous motion view. In basketball and football, while the advantage of high-frame-rate modelling is smaller, it remains notable. This is likely because the model can leverage additional contextual cues, such as player behaviour, audience reactions, or score changes, to infer outcomes even in the absence of keyframes. For instance, whether a shot was successful or a pass occurred.

Beyond QA, our model also demonstrates strong high-frame-rate captioning capabilities. Even advanced models like GPT-4 struggle with accurate captioning of high-speed sports videos. Appendix C presents a comparison of sports captioning outputs across different models, showcasing the effectiveness of our approach.

### 5.3. Variable-Frame-Rate Testing

In high-frame-rate modeling, since each frame needs to be encoded using the encoder, this can significantly impact the model's inference speed. Testing at a frame rate lower than the training frame rate can increase speed with minimal performance loss. Fig. 3 shows the performance and inference time of variable-frame-rate testing. The inference is considerably slowed down by the image encoder when the test FPS increases, as Fig. 3a shows. Especially when outputting only a very small number of tokens, such as when answering questions, the time taken by the image encoder in high-frame-rate modelling even surpasses that of the backbone LLM.

Variable-frame-rate decoding proves to be an effective solution for addressing this challenge. As shown in Fig. 3b, adapting the high-frame-rate model to low-frame-rate inference achieves comparable performance on generic video benchmarks like Video-MME, with only a slight degradation compared to evaluation at FPS = 16. Moreover, it performs favourably against models trained at the same test FPS. This flexibility allows developers to balance computational efficiency and high-frame-rate advantages based on specific tasks. For example, lower FPS can be used for fast reasoning in general video comprehension, while higher FPS is preferable for fine-grained understanding in high-speed video analysis tasks.

### 5.4. Analysis on High-Frame-Rate Aligner

To gain deeper insights into the mechanism of the high-frame-rate aligner, we first analyze the visual features output by the image encoder to verify the presence of subtle

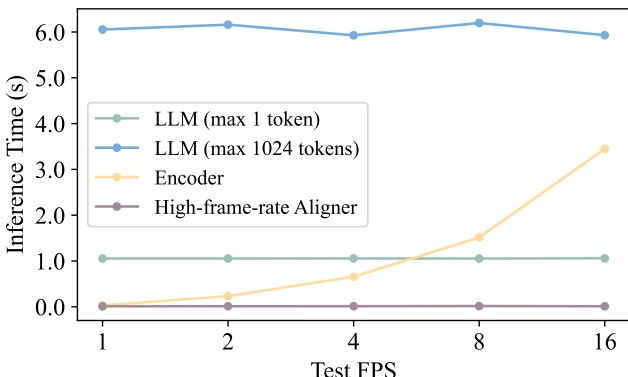

(a) Time consumption of different modules of F-16 when inference. Models are prompted to do video captioning on Video-MME Long set (with 300 videos) with different test FPS and different max lengths.

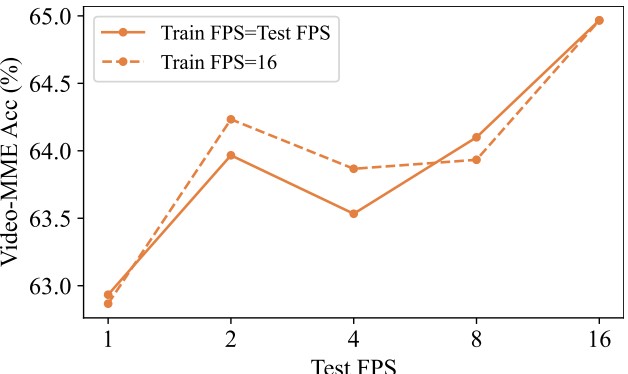

(b) Video-MME performance of models trained and tested at different FPS. The solid line indicates that the model is trained and tested using the same FPS, while the dashed line indicates the model is trained at FPS = 16 and tested at a lower FPS.

*Figure 3.* Performance and inference time analysis of variable-frame-rate testing.

high-frame-rate details. We compare the cosine similarity $d_{\text{cos,before}}$ between visual features across frames, as shown in Table 3. Additionally, we apply a $2 \times 2$ Max2DPool$(\cdot)$ function to the visual features and compute the cosine similarity after pooling, denoted as $d_{\text{cos,after}}$, to observe the impact of this basic compression method on the feature representations. As expected, cosine similarity before pooling decreases progressively as frames shift over time, eventually dropping to 0.5898 by the 4th frame, indicating clear differences in visual features. However, after max pooling, the cosine similarity remains high (0.8828) at Frame 4, despite significant changes in the scene and player actions. This suggests that fine-grained visual features are not dominant compared to global features and are largely suppressed by max pooling. Based on this observation, we hypothesize that preserving the complete visual features as input to the high-frame-rate aligner can lead to significant performance improvements, as it retains fine-grained temporal details

*Table 3.* The average cosine similarity of all image tokens between the frames and the reference frame. $d_{\cos,\text{before}}$ refers to the cosine distance before pooling, while $d_{\cos,\text{after}}$ refers to the cosine distance after pooling.

| Reference Frame | Frame 1 | Frame 2 | Frame 3 | Frame 4 |
|---|---|---|---|---|
|  |  |  |  |  |
| $d_{\cos,\text{before}}$ | 0.8945 | 0.7656 | 0.6680 | 0.5898 |
| $d_{\cos,\text{after}}$ | 0.9609 | 0.9180 | 0.9023 | 0.8828 |

*Table 4.* Results on general video data using different pooling strategies for models at different frame rates.

| FPS | Pooling | Video-MME | | | |
|---|---|---|---|---|---|
| | | Avg.↑ | S↑ | M↑ | L↑ |
| 1 | pre | 62.3 | 76.6 | 59.7 | 50.6 |
| 1 | post | 62.9 | 77.7 | 60.3 | 50.8 |
| 16 | pre | 60.8 | 75.0 | 58.2 | 49.2 |
| 16 | post | **65.0** | **78.9** | **63.2** | **52.8** |

essential for high-speed video understanding.

We conduct experiments on general video data using different pooling strategies across models with varying frame rates. Specifically, we examine pre-pooling, where max pooling is applied before the modality aligner, and post-pooling, where max pooling is applied after the modality aligner. The results are presented in Table 4. For the FPS= 1 model, post-pooling provides only a marginal improvement of less than 1% on the Video-MME benchmark. However, for the FPS= 16 model, pre-pooling proves even less effective than in the FPS= 1 scenario, whereas post-pooling yields a notable 4.2% improvement. These findings suggest that the high-frame-rate aligner heavily relies on inter-frame variations in the visual features to learn effectively. Pre-pooling prematurely removes fine-grained temporal details, making it harder for the high-frame-rate aligner, with its larger parameter set, to extract complex motion patterns, potentially leading to overfitting and degraded performance. In contrast, post-pooling preserves these temporal details, thereby enhancing the model's overall performance.

Additionally, we conducted another way to reduce the test FPS to see how the high-frame-rate aligner works. The parameters $\mathbf{W_P} \in \mathcal{R}^{wd \times wh}, \mathbf{b_P} \in \mathcal{R}^{wh}, \mathbf{W_Q} \in \mathcal{R}^{wh \times h}, \mathbf{b_Q} \in \mathcal{R}^{h}$ of the aligner are trimmed to submatrices according to the test FPS $s$, as in Eqns. (10)-(11):

$$\mathbf{W'_P} = \mathbf{W_P}[: sd, : sh], \ \mathbf{b'_P} = \mathbf{b'_P}[: sh] \quad (10)$$

$$\mathbf{W'_Q} = \mathbf{W_Q}[: sh, :], \ \mathbf{b'_Q} = \mathbf{b_Q}. \quad (11)$$

This trimming approach effectively leverages the local parameters of the high-frame-rate aligner for low-frame-rate decoding, distinguishing it from the repeating-frame method introduced in Section 3.3. To evaluate its effectiveness, we compare trimming with frame repetition for low-frame-rate inference, denoted as "Trimming" and "Repeat", respectively. The results, presented in Table 5, highlight the differences between these two methods.

*Table 5.* Variable-frame-rate decoding results on Video-MME using different methods. "Trimming" represents only using the submatrices of the original aligner, and "Repeat" represents repeating frames to meet the input dimension of the original aligner. The normal decoding results are also provided.

| Method | FPS | Video-MME | | | |
|---|---|---|---|---|---|
| | | Avg.↑ | S↑ | M↑ | L↑ |
| - | 16 | 65.0 | 78.9 | 63.2 | 52.8 |
| Trimming | 8 | 62.3 | 76.7 | 61.0 | 49.2 |
| Repeat | 8 | 63.9 | 78.0 | 62.4 | 51.4 |

The "Trimming" decoding method results in a performance decline, whereas the "Repeat" method nearly preserves the high-frame-rate performance. This suggests that the high-frame-rate aligner does not simply perform an averaging operation in the feature space, despite being initialized that way. Instead, it effectively extracts motion information between semantic features through its 3-layer MLP structure, rather than relying solely on local parameter adjustments.

Additionally, we experimented with more structures for the high-frame-rate aligner, but the results were unsatisfactory. This further confirms that using linear mappings or MLPs as aligners is more effective in preserving the semantic space of visual features. Detailed results of these experiments are provided in Appendix B.

## 6. Conclusions

This work presents F-16, a powerful video LLM designed for 16 FPS video understanding, along with the high-frame-rate aligner. To the best of our knowledge, this is the first

study to explore high-frame-rate processing in video LLMs. We also introduce a method to initialize video LLMs from pre-trained image LLMs, enabling F-16 to leverage their rich pre-trained knowledge for enhanced video understanding. Additionally, we propose a variable-frame-rate decoding method, which balances high-frame-rate performance and inference cost without requiring additional training. As a result, F-16 achieves SOTA performance in general video QA among models of similar size and demonstrates a clear advantage in high-frame-rate video understanding, outperforming commercial models such as GPT-4o. This work opens new directions for advancing high-frame-rate video comprehension in multimodal LLM research.

## Impact Statement

This paper presents work whose goal is to advance the field of Machine Learning. There are many potential societal consequences of our work, none of which we feel must be specifically highlighted here.

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

# A. Examples of the Sports Video Data

*Table 6.* Examples of sports data, including Gym, Diving, Soccer and NBA.

| Sports Type | Category | Content |
|---|---|---|
| **Gym** | QA | **Q**: Please describe the gym video between 0.0s and 5.7s in detail. **A**: The player performs "round-off, flic-flac on, stretched salto backward with 1 turn off". |
| | Caption | **Q**: Please describe the gym video in detail and pay attention to the time. **A**: The video is a sport video about Floor Exercise. The player first performs "double salto backward piked" between 19.7s and 21.5s. Then, the player performs "2 turn or more with heel of free leg forward at horizontal throughout turn" between 28.0s and 29.4s. Then, the player performs "salto backward stretched with 2.5 twist" between 43.3s and 44.3s. Then, the player performs "switch leap to ring position" between 51.2s and 52.1s. Then, the player performs "split leap with 1 turn" between 52.9s and 53.9s. Then, the player performs "salto forward tucked" between 64.1s and 64.9s. Then, the player performs "salto backward stretched with 2 twist" between 66.2s and 67.9s. Then, the player performs "switch leap with 1 turn" between 71.3s and 72.5s. Then, the player performs "double salto backward tucked" between 91.0s and 93.0s. |
| **Diving** | QA | **Q**: What type of take off did the diver use? **A**: Reverse. |
| | Caption | **Q**: Please describe this diving video. **A**: The diver executed a back take-off, transitioned into a 15 somersaults, and incorporated a 5 twist while in a free position. |
| **Soccer** | QA | **Q**: How many football passes are there in the video? **A**: 2. |
| **NBA** | QA | **Q**: Did the shot go in? **A**: Yes. |
| | Caption | **Q**: Explain the NBA video clip thoroughly, ensuring fleeting details are captured. **A**: Hawks player No. 2 Wallace tossed the ball to No. 1 Johnson on the left side of the three-point line. Johnson jumped up from the left side of the basket and caught the ball with his right hand. That's when the Celtics' No. 7 tried to stop him. Johnson made two center-foot stance-up moves and made a feint pass to the No. 10 from beyond the three-point line. At 11:45, No. 10 made a three-point shot in front left of the rim before being blocked by Celtics player No. 0 Tatum. After being blocked, the number 10 grabs the ball in the same position and dribbles once, passing the ball to the number 1 on the left baseline of the frame. |

Table. 6 shows the examples for each category of the sports video used in training and evaluation that are mentioned in Sec. 4.2.

# B. Different Modality Aligners for High-Frame-Rate Understanding

The high-frame-rate aligner mentioned in Sec. 3 uses linear layers that compress the frames temporally as the main layers and form an MLP. However, more structures are tried in our experiments, as shown in Table. 7-9. Linear is finally selected based on the experimental results.

*Table 7.* Comparison between linear and CNN methods.

| Structure | FPS | Max Time | LLM Training | Video-MME | | | |
|---|---|---|---|---|---|---|---|
| | | | | **Avg.↑** | **S↑** | **M↑** | **L↑** |
| Linear | 32 | 30 | LoRA | 61.5 | 75.0 | 59.2 | 50.3 |
| 2D-CNN | 32 | 30 | LoRA | 57.7 | 69.1 | 55.9 | 48.2 |
| 3D-CNN | 32 | 30 | LoRA | 57.2 | 70.0 | 54.8 | 46.9 |

Table. 7 shows the comparison between linear and CNN methods, which implement spatiotemporal convolution to replace the linear layer and the pooling layer, in order to make the model compress and understand spatiotemporal information together. 2D-CNN conducts convolution on the temporal dimension and spatial visual token dimension together, while 3D-CNN resizes the spatial dimension into a 2D graph like the operation before pooling in Sec. 3. However, the CNN models fail to understand fine-grained video details and achieve worse final results. The results show that spatiotemporal

compression and encoding are difficult to conduct in one step. Besides, the CNN models get quite a different structure compared to the aligner in image models, meaning that it cannot be easily initialized with the original pretrained aligner and requires much more training costs to conduct image pretraining.

*Table 8.* Comparison between linear and dual linear method.

| Structure | FPS | Max Time | LLM Training | Video-MME | | | |
|---|---|---|---|---|---|---|---|
| | | | | Avg.↑ | S↑ | M↑ | L↑ |
| Linear | 32 | 30 | LoRA | 61.5 | 75.0 | 59.2 | 50.3 |
| Dual Linear | 32 | 30 | LoRA | 58.8 | 72.3 | 55.8 | 48.2 |

Table. 8 shows the comparison between linear and dual linear methods. Dual linear replaces the original max pooling layer with a learnable linear layer, aiming to learn better spatial compression. Though the model performs better in training loss, it performs worse in the test stage, showing worse generalization ability. That's probably due to the limitation in the scale of video data, making the learned spatial compression linear slightly overfits the training set. Meanwhile, the spatial linear layer is difficult to initialize with the modules in image LLMs or human-tuned parameters, leading to the waste of the rich knowledge in image LLMs.

*Table 9.* Comparison between linear and attention method.

| Structure | FPS | Max Time | LLM Training | Video-MME | | | |
|---|---|---|---|---|---|---|---|
| | | | | Avg.↑ | S↑ | M↑ | L↑ |
| Linear | 32 | 30 | LoRA | 61.5 | 75.0 | 59.2 | 50.3 |
| Attention | 32 | 30 | LoRA | 61.8 | 74.2 | 60.3 | 50.9 |
| Linear | 16 | 60 | Full | 63.5 | 78.1 | 60.6 | 51.8 |
| Attention | 16 | 60 | Full | 62.9 | 76.8 | 60.2 | 51.9 |

Table. 8 shows the comparison between linear and attention methods. Attention uses similar structures to linear, but replaces the first linear layer with a self-attention layer, to further catch the connection of the frames. It is observed that attention does achieve better results when training LoRA only, but fails to maintain the advantage when training full LLM. The difference is probably brought by the minimal transform of attention that fits the original pretrained LLM better (each frame is independently processed with $W_V$ matrix in self-attention) but does not bring more advantages when all the parameters are trainable. Meanwhile, the quadratic complexity of the attention mechanism also makes it harder to extend to a higher max time. In our implementation, due to the VRAM limitation, it cannot be extended to 110s with 16 FPS like the linear one, which also reduces the potential of attention aligner.

## C. Captioning Cases for High-Speed Sport Videos

We show detailed NBA captions generated by models trained at different frame rates in Table 10. The groundtruth caption and GPT-4o's caption are also provided. GPT-4o and the FPS = 1 model misses ball-passing information or describes it incorrectly, while the high-frame-rate model is able to catch these details.

## D. Visualization of Sports Videos at Different FPS

We randomly sample a video from Diving48 and show the sampled frames at different FPS in Fig. 4-6.

*Table 10.* Captioning Cases for a 10-second NBA video of different models.

| Text Type | Content |
|---|---|
| Groundtruth | Jazz player No. 23 held the ball outside the left three-point line, guarded by Lakers player No. 46. He suddenly passed the ball to Jazz player No. 40, a teammate cutting into the paint from the three-point line in the middle. The latter attempted a layup at 10:25, and Lakers player No. 3 raised his right hand, trying to block the shot. The ball didn't go in. Lakers player No. 15 grabbed the rebound and passed it to player Lakers player No. 3. Lakers player No. 3 then dribbled the ball past half-court. |
| GPT-4o (Inputted with 120 frames) | The NBA video clip captures a sequence from a game between the Utah Jazz and an opposing team. The play begins with the Jazz on offense, setting up a play on the perimeter. The player with the ball attempts to drive towards the basket but is met with strong defense from the opposing team. The ball handler then passes to a teammate who is positioned near the three-point line. This player makes a move to penetrate the defense but is also challenged by defenders. The ball is then passed around the perimeter as the Jazz look for an open shot. Eventually, a player attempts a shot near the basket but misses. The opposing team secures the rebound and quickly transitions into a fast break, pushing the ball up the court. The clip ends with the opposing team advancing towards their offensive end, looking to capitalize on the transition opportunity. |
| Generated by model trained at FPS = 1 | The No. 17 player of the Jazz held the ball in the backcourt. He dribbled once with his left hand, then switched to his right hand, and moved towards the frontcourt. After three more dribbles, he entered the paint. Then he jumped up and threw the ball towards the basket at 12:24. The basketball hit the rim and bounced out. The No. 5 player of the Lakers jumped up and grabbed the rebound with both hands. |
| Generated by model trained at FPS = 16 | Jazz player No. 24 dribbled the ball once outside the three-point line on the left side of the basket and passed it to Jazz player No. 15 who moved to the right side of the basket outside the three-point line. Jazz player No. 15 took a jump shot at 10:23, but failed to score. Lakers player No. 3 jumped up and raised his right hand to try to block the shot. The basketball hit the rim and bounced out. Then Jazz player No. 0 grabbed the rebound and passed the ball to Jazz player No. 24 who moved to the right side of the basket outside the three-point line. |

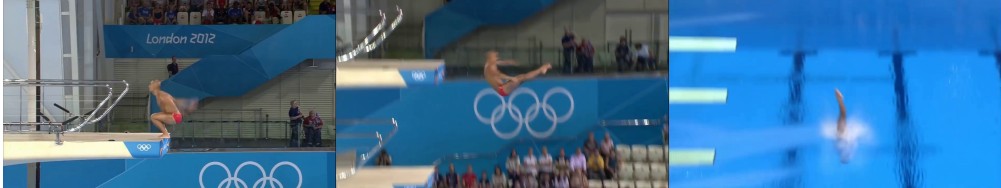

*Figure 4.* Visualization at FPS=1

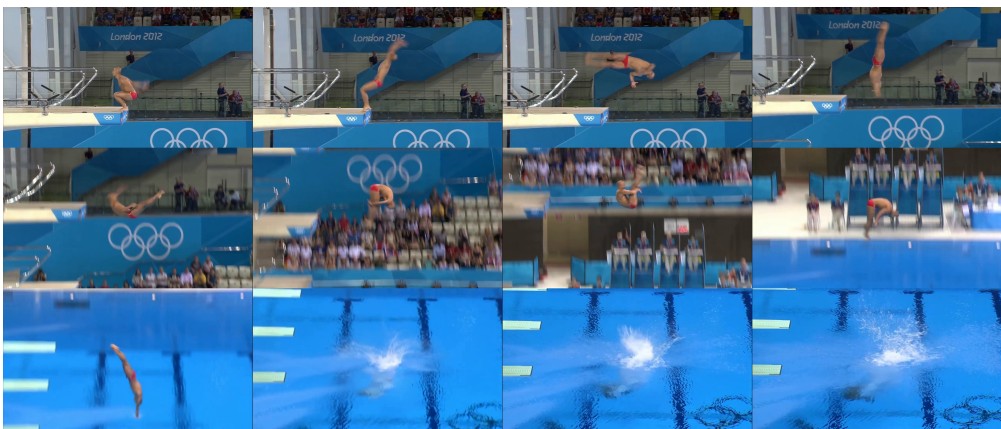

*Figure 5.* Visualization at FPS=4

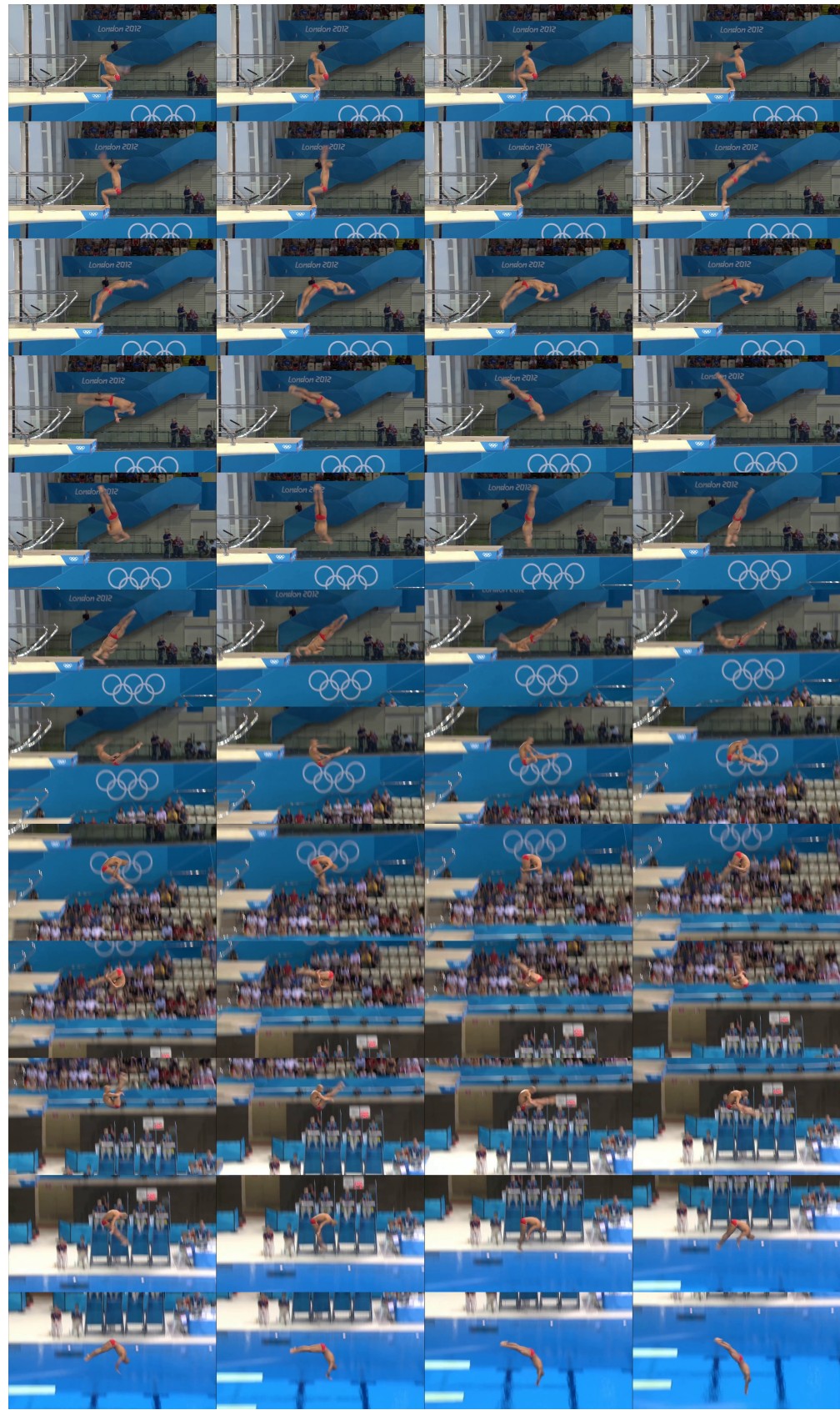

*Figure 6.* Visualization at FPS=16

