# OpenReview forum: "Improving LLM Video Understanding with 16 Frames Per Second"
_ICML.cc/2025/Conference — ICML 2025 poster_

### Official Review · Reviewer_g4TB · 2025-03-10

**Overall Recommendation:** 2

**Summary:**

The paper explores high FPS in video understanding with MLLM. It is an interesting and meaningful attempt and the authors employ some techniques to solve the problem of excessive number of tokens. The performance gain is promising in some specialized scenarios like sports as expected.

## update after rebuttal

I appreciate the author's response, but I still have concern on the temporal design of pre-fusing 16 or other number of frames within one second. The current temporal design equals to decoupled visual temporal perception module with a LLM. The utilization of high frame rate input is restricted by the temporal fusion module. A smarter token selection rather than fusion enables more systematic VLM for processing high frame rate videos.

**Claims And Evidence:**

The claims are supported by the experimental results.

**Essential References Not Discussed:**

No missing references

**Experimental Designs Or Analyses:**

More analysis on the aligner is desired, including input temporal range, output number of tokens, inserting positions, etc.

**Methods And Evaluation Criteria:**

1. The method of merging high frame-rate tokens is very simple, utilizing the generalization from single-frame to multiple-frame scenarios. However, this strategy has limited temporal capacity and cannot fully leverage the temporal dynamics in the high frame-rate input, slightly superior to average pooling. Either heuristic token preprocessing or more systematic model design is expected.
2. The evaluation on existing general video benchmarks cannot showcase the advantage of high frame-rate training, so the authors extend to the high-speed sports scenarios. It is necessary to supplement more scenarios beyond sports, compare with more video specialist models and conduct more comprehensive ablations on the high frame-rate token processing.

**Other Comments Or Suggestions:**

None

**Other Strengths And Weaknesses:**

The problem is meaningful but the authors should study more technical designs on the aligner and conduct more comprehensive experiments.

**Questions For Authors:**

None

**Relation To Broader Scientific Literature:**

The high frame rate exploration is a significant part that pushes VLM to achieve human-level perception, which can be related with human perception literatures.

**Theoretical Claims:**

Correct

---

> ### Author Rebuttal · Authors · 2025-03-31
>
> Thank you very much for your thoughtful and constructive feedback on our paper. Below, we will respond to the questions you have raised.
>
> ---
>
> 1: **The evaluation on existing general video benchmarks cannot showcase the advantage of high frame-rate training. & Supplement more scenarios beyond sports**
>
> In fact, in general video understanding, F-16 also demonstrates significant advantages in short video benchmarks. Note that F-16 uses less data (less video data and image data) compared to most of the models in Table 1, and it only uses LLaVA-Video-178k. Besides, as for more scenarios, we also evaluate F-16 on a very recent benchmark, FAVOR-Bench, which targets evaluating **fine-grained video motion understanding**. F-16 achieves SOTA results  on it among 7B models.
>
> We list the results on these benchmarks here to more clearly present the advantages of F-16:
>
> |                  | Video-MME Short | NExT-QA            | TemporalBench   | MotionBench     | FAVOR-Bench        |
> | ---------------- | --------------- | ------------------ | --------------- | --------------- | ------------------ |
> | Previous 7B SOTA | 75.7 (NVILA)    | 83.2 (LLaVA-Video) | 24.7 (Qwen2-VL) | 52.0 (Qwen2-VL) | 41.5 (VideoLLaMA3) |
> | F-16             | **78.9**        | **84.1**           | **37.2**        | **54.5**        | **46.0**           |
>
> For long video understanding, although F-16 is not outstanding, it still shows competitive results. This is because only when processing short videos, the frame sampling is done at FPS = 16, matching the training situation. For long videos, the FPS is much lower than 16. For instance, as F-16 samples a maximum of $110\times 16=1760$ Frames, it perceives a 1760-second video at FPS=1. Sparse frame sampling will lead to slightly poorer performance of the high-frame-rate aligner, resulting in average results for long videos.
>
> ---
>
> **2: More technical designs and experiments on the aligner.**
>
> We studied various high-frame-rate modeling structures. All attempts confirmed that visual-language alignment via linear transformations helps prevent model performance decline, which means linear projections effectively and efficiently align the visual encoder's output semantic space with the LLM's input space. This is consistent with prior work like NVILA.
>
> Appendix B details the studied structures. Using CNN modules to capture frame spatiotemporal differences led to worse results than MLP, suggesting CNN aligners may harm semantic information derived from the visual encoder. Replacing max pooling with a learnable linear layer improved training loss but is worse when testing. Using a self-attention layer to replace the MLP projector's first linear layer to extract frame dynamic changes also affected visual feature semantics, resulting in slightly worse performance when scaling up the training parameters.
>
> Ultimately, we chose a 2-layer MLP structure. Its first linear layer ensures semantic alignment between LLM input and visual encoder output spaces, and the second compresses duplicated information in continuous frames. It is well-known that any continuous mapping function can be represented arbitrarily accurately by a 2-layer MLP with sufficiently large hidden layer dimension, which provides an insight into our motivation. Experimental results also validate this design.
>
> ---
>
> **3: More analysis on the aligner is desired, including input temporal range, output number of tokens, inserting positions, etc.**
>
> As for the temporal range, we have trained models at different FPS, and set the width $w$ of the processing window equal to the FPS. As the input FPS gradually increases from 1 to 16, the performance of the model shows an upward trend, shown in Fig. 3(b) "Train FPS=Test FPS". Besides training, testing at different FPS is also tried, shown in Fig. 3(b) "Train FPS=16".
>
> Though the input number of tokens to the aligner increases with FPS, the output number of tokens remains the same due to the processing window width of the aligner remains the same with FPS. This setting enables a fair comparison of models with different frame rates.
>
> For inserting positions, since the high-frame-rate aligner is a replacement to the single-frame aligner, only pre- and post-pooling used by other VLLMs are evaluated. Table 3 shows pre-pooling causes larger differences between adjacent frames, and Table 4 shows post-pooling performs much better under high-frame-rate settings. These indicate that the high-frame-rate aligner heavily relies on inter-frame variations in visual features for effective learning.  If adjacent sampled frames are no longer similar to each other, the high-frame-rate aligner will struggle to learn well.
>
> ---
>
> **4: Present the visualizations of model inputs under different frame rates.**
>
> We visualize videos at different frame rates here: https://github.com/F-16-LLM/Rebuttal/blob/main/README.md
>
> Under low-frame-rate cases like FPS = 1, many details will be missed. We will add this part to the updated paper.

---

### Official Review · Reviewer_xf2Z · 2025-03-14

**Overall Recommendation:** 4

**Summary:**

This paper studies the problem of high-frame-rate video understanding. The authors claimed that existing methods for video understanding merely sample video frames at a low FPS (mostly lower than 2), where there exists critical information loss. To tackle this problem, they introduce F-16, a novel multimodal large language model (MLLM) specially designed for high-frame-rate video understanding at 16 FPS. The main contributions are summarized as: 1) [model]: The first high-frame-rate video LLM, 2) [Benchmark]: A High-frame-rate sports video benchmark, and 3) [method]: Efficient variable-frame-rate decoding.

**Claims And Evidence:**

The main claim of the paper is that "the existing paradigm for video understanding (sampling at around 2FPS) is sub-optimal, which would lose much information when facing highly dynamic videos". This claim is clear, reasonable, and supported with convincing evidence.

**Essential References Not Discussed:**

No

**Experimental Designs Or Analyses:**

The experiments are conducted on both the proposed high-frame-rate benchmark (NBA videos) with manual annotations and public video understanding benchmarks. The experiment protocols are reasonable.

**Methods And Evaluation Criteria:**

Yes

**Other Comments Or Suggestions:**

N/A

**Other Strengths And Weaknesses:**

Generally, this is a good paper about LLM-based video understanding. Extending existing paradigms to 16FPS is a non-trivial setting. The proposed high-frame-rate aligner can well-balance the efficiency and performance.

**Questions For Authors:**

N/A

**Relation To Broader Scientific Literature:**

The key contribution is to extend the existing Video-LLMs from a low frame rate to a high frame rate, which is more natural and more suitable for analyzing highly dynamic videos. To my best knowledge, this is the first work to do such exploration.

**Theoretical Claims:**

There are no proofs in the submission.

---

> ### Author Rebuttal · Authors · 2025-03-31
>
> Thank you for the positive rating for the paper. We sincerely appreciate your recognition of our work.

---

> > ### Comment · Reviewer_xf2Z · 2025-04-02
> >
> > Thanks for the response from the authors. I'm keeping my original rating.

---

### Official Review · Reviewer_nuva · 2025-03-16

**Overall Recommendation:** 3

**Summary:**

This paper proposes a new method F-16 that increases the frame rate of existing video LLM to 16 frames per second (FPS). The paper argues that existing video LLMs, which typically operate on low frame rates (e.g., 1 FPS), lose crucial dynamic visual information. F-16 aims to address this by processing videos at a significantly higher frame rate while employing a visual-text aligner to compress redundant visual information within 1-second clips. This allows the model to capture subtle but important motion cues. The paper claims that F-16 achieves state-of-the-art performance among 7B parameter video LLMs on both general (Video-MME, TemporalBench) and fine-grained video understanding benchmarks. Further, it excels in complex spatiotemporal tasks such as high-speed sports analysis, outperforming models like GPT-4o and Gemini 1.5 Pro. The authors also propose a novel decoding method enabling efficient low-frame-rate inference without retraining the model.

**Claims And Evidence:**

The claims made in the submission are generally well-supported by evidence.

Claim: Higher frame rates enhance video understanding. This is supported by the consistent performance gains observed across various benchmarks (Video-MME, TemporalBench, MotionBench, sports datasets) when comparing F-16 with other 7B models.

Claim: F-16 excels in high-speed sports analysis. Table 2 demonstrates a significant performance advantage of F-16 in tasks like gymnastics, diving, basketball, and football compared to other video LLMs. The comparison to GPT-4o and Gemini 1.5 Pro is also compelling.

**Essential References Not Discussed:**

No

**Experimental Designs Or Analyses:**

The experimental designs and analyses seem sound.

Comparison with other models: The paper compares F-16 with several state-of-the-art video LLMs and proprietary models (GPT-4o, Gemini 1.5 Pro) to demonstrate its effectiveness.

Ablation studies: The ablation studies on pooling strategies and alignment strategies (Table 4 and Table 5) provide insights into the importance of different components of F-16.

Analysis of high-frame-rate aligner: The analysis of visual features output by the image encoder and the cosine similarity analysis helps to understand the benefits of the high-frame-rate aligner.

**Methods And Evaluation Criteria:**

The proposed methods and evaluation criteria seem reasonable for the problem of video understanding.

High-frame-rate sampling: The decision to use 16 FPS is empirically motivated and aligns with the need to capture more dynamic visual information. The claim that this is a better trade-off is reasonable.

Visual-text aligner with 3-layer MLP: The proposed aligner is designed to transform visual features to text like tokens. The MLP design is justified for its efficacy and compatibility with existing image encoders.

Evaluation benchmarks: The use of standard video understanding benchmarks like Video-MME, TemporalBench, MotionBench, and sports datasets is appropriate for evaluating the performance of F-16. The choice of metrics (Accuracy, F1-score) is also standard for the tasks considered.

**Other Comments Or Suggestions:**

The paper could benefit from a more detailed analysis of the computational cost of F-16, including the memory footprint and inference time.

It would be interesting to explore the potential of using even higher frame rates (e.g., 30 FPS) and the trade-offs between performance and computational cost.

**Other Strengths And Weaknesses:**

Strengths:

-Novelty: The focus on high-frame-rate video understanding is a novel and important direction for video LLMs.

-Performance: F-16 achieves state-of-the-art performance among 7B models.

-Efficiency: The variable-frame-rate decoding method enables efficient inference without retraining.

Completeness: The paper is well-written and provides sufficient details about the proposed methods and experimental results.

Weaknesses:

While variable frame rate decoding is discussed, the actual cost and memory footprint requirements are not shown, this is a limiting factor.

**Questions For Authors:**

Can you provide more detailed information about the training procedure, including hyperparameter settings, training time, and the resources (e.g., number of GPUs) used for training? This information would help to better assess the reproducibility of the results.

If the training process is relatively efficient, it would make the paper more appealing to the wider community.

What are the computational costs (memory footprint, inference time) of F-16 compared to other video LLMs? This information is important to understand the practical limitations of the proposed approach.

If F-16 has a manageable computational cost, it would strengthen the paper's claim of being a practical solution for high-frame-rate video understanding.

**Relation To Broader Scientific Literature:**

The paper builds upon the existing literature on video LLMs, particularly those that focus on improving video understanding by leveraging pre-trained image encoders and modality alignment techniques. The key contribution of this paper is the emphasis on higher frame rates, which has been relatively unexplored in the context of LLMs. The paper acknowledges and compares its approach with existing methods for video processing and temporal input compression.

**Theoretical Claims:**

There are no theoretical claims in the paper that require proof checking.

---

> ### Author Rebuttal · Authors · 2025-03-31
>
> Thank you very much for your thoughtful and constructive feedback on our paper. Below, we will respond to the questions you have raised.
>
> ---
>
> **1: Can you provide more detailed information about the training procedure, including hyperparameter settings, training time, and the resources (e.g., number of GPUs) used for training?**
>
> Regarding the training procedure, F-16 is first initialized from LLaVA-OneVision-7B, as shown in Sec. 4.1. Then, F-16 is trained on general videos. At this stage, the image encoder stays frozen while other parts of the model are updated. To further verify the advantages of high-frame-rate modeling, the model is then fine-tuned on high-speed sports video. LoRA is adapted to the LLM and serves as the only trainable module in this stage.
>
> As for other training settings, F-16 is trained with Adam optimizer and cosine scheduler. The learning rate is set to 2e-5, the warm-up ratio is set to 0.03, and the batch size per device is 1. For general video training, 128 H100 GPUs are used for training for 1 epoch (about 13000 steps of updates), which takes about 35 hours. As for high-speed sports fine-tuning, 64 H100 GPUs are used for training 5 epochs (about 9000 steps of updates), which takes about 20 hours. For comparison, the FPS=1 model takes about 18 hours for general video training and 10 hours for high-speed sports fine-tuning.
>
> Note that the difference in training time primarily comes from two parts: a small portion is due to the increase in encoder duration, while the major part is the catastrophic growth in CPU time when reading more frames in long videos, which leads to less GPU usage. We have not optimized video frame extraction, and just use the Python Library"Decord" for frame extraction. Appropriate optimization or preprocessing can significantly improve training speed, such as extracting frames in advance.
>
> ---
>
> **2: What are the computational costs (memory footprint, inference time) of F-16 compared to other video LLMs? This information is important to understand the practical limitations of the proposed approach.**
>
> With the high-frame-rate aligner, though 16 times more visual frames are inputted, the tokens into the LLM remain the same as the low-frame-rate model. Therefore, the computational cost for LLM is the same as compared of the low-frame-rate one. The difference mainly comes from the encoder and the aligner.
> - Inference time. As compared in Fig. 3(a), the inference time of the proposed high-frame-aligner can be ignored compared with the LLM and the visual encoder, and the comparison to other models can mainly focus on the visual encoder. To our observation, the inference time for the visual encoder grows linearly with the input frame time. Therefore, the inference time of the visual encoder increases by 16x, which makes the total inference time of F-16 1x longer than that of the comparable video LLM with FPS=1.
> - Memory footprint. The memory cost can be divided into 4 parts: the final visual tokens, the hidden layer of the aligner, the output of the visual encoder, and the inner computation cost of the visual encoder. The final visual tokens are kept at the same number, which does not increase in memory cost. For the other 3 parts, though it seems that they will grow linearly along with the number of frames, they can be handled sequentially because of the independence between different processing windows. For instance, using a for-loop to process frames between processing windows. Therefore, there is no significant increase in memory cost observed from F-16.

---

### Decision · Program_Chairs · 2025-05-01

**Decision:**

Accept (poster)

**Comment:**

The manuscript received ratings of 4, 3, and 2. Reviewers generally appreciated the idea, impressive performance of the proposed approach while maintaining the efficiency, and papre writing. Reviewers also raised several questions in the initial review including, additional details about the training procedure (e.g., hyperparameter settings, training time, and the resources),  input temporal range, and output number of tokens. Post-rebuttal, two reviewers remained positive while the third reviewer still has concerns on the temporal design of pre-fusing a number of frames. Authors presented the required additional details in the rebuttal, including the training procedure such as, hyperparameter settings, training time, and the resources (e.g., number of GPUs) used for model training. Given the reviews and the rebuttal, the AC agrees with reviewer nuva and reviewer xf2Z that the main contribution of extending the existing video-LLMs from a low frame rate to a high frame rate is interesting and is well supported with reasonable experimental protocols, demonstrating consistent performance gains observed across various benchmarks. The altnertaive idea of exploring a smarter token selection, as suggested by reviewer g4TB is also worth exploring, but the authors showed that their proposed approach based on high FPS in video understanding with MLLM while employing a visual-text aligner to compress redundant visual information, is simple and leads to performance gains (also acknowledged by reviewers).